# Identification of the governing equation of stimulus-response data for run-and-tumble dynamics

**Shicong Lei**[1,2☯], **Yu'an Li**[2,3,4☯], **Zheng Ma**[1,2], **Hepeng Zhang**[2,3,4*], **Min Tang**[1,2,4*]

**1** School of Mathematics, Shanghai Jiao Tong University, Shanghai, China, **2** Institute of Natural Sciences, Shanghai Jiao Tong University, Shanghai, China, **3** School of Physics and Astronomy, Shanghai Jiao Tong University, Shanghai, China, **4** Key Lab of Scientific and Engineering Computing, Ministry of Education, Shanghai Jiao Tong University, Shanghai, China

☯ These authors contributed equally to this work.
* hepeng_zhang@sjtu.edu.cn (HZ); tangmin@sjtu.edu.cn (MT)

**Data availability statement:** The code and data required to replicate findings reported in the article are available at https://github.com/ leiemily/NNsummary.

## Abstract

The run-and-tumble behavior is a simple yet powerful mechanism that enables microorganisms to efficiently navigate and adapt to their environment. These organisms run and tumble alternately, with transition rates modulated by intracellular chemical concentration. We introduce a neural network-based model capable of identifying the governing equations underlying run-and-tumble dynamics. This model accommodates the nonlinear functions describing movement responses to intracellular biochemical reactions by integrating the general structure of ODEs that represent these reactions, without requiring explicit reconstruction of the reaction mechanisms. It is trained on datasets of measured responses to simple, controllable signals. The resulting model is capable of predicting movement responses in more realistic, complex, temporally varying environments. Moreover, the model can be used to deduce the underlying structure of hidden intracellular biochemical dynamics. We have successfully tested the validity of the identified equations based on various models of *Escherichia coli* chemotaxis, demonstrating efficacy even in the presence of noisy measurements. Moreover, we have identified the governing equation of the photo-response of *Euglena gracilis* cells using experimental data, which was previously unknown, and predicted the potential architecture of the intracellular photo-response pathways for these cells.

## Author summary

Microscopic organisms like bacteria and algae often adjust their movement in response to changing environments, such as light or chemical signals. They do this using a behavior called run-and-tumble, alternating between straight swimming and reorientation. Understanding how internal cell processes drive this behavior is difficult, especially when

**Funding:** M.T. is supported by National Natural Sciences Fundation of China (Grants No. NSFC12031013). H. P. Z. is supported by the National Natural Science Foundation of China (Grants No. 12225410 and No. 12074243), the Ministry of Science and Technology Most China (Grant No. 2021YFA0910700). The sponsors or funders play no role in the study design, data collection and analysis, decision to publish, or preparation of the manuscript.

we can't directly observe the biochemical pathways involved. In this study, we present a machine learning approach that discovers the governing rules behind these behaviors using only observable input-output data. Our model, based on neural networks, learns from how cells respond to simple stimuli and can predict how they behave under more complex, realistic conditions, without needing to know the details of internal reactions. We validated the method using simulations of bacterial chemotaxis and real experimental data from *Euglena gracilis*, a microorganism that responds to light. Our model accurately predicted cell responses and revealed insights into possible internal signaling structures. This approach provides a powerful tool for studying biological behavior and could help uncover how other organisms process environmental information.

## Introduction

In the intricate natural environment, organisms have developed a variety of sensory receptors to perceive external stimuli, including chemoattractants [1–13], thermal changes [14], hydro-dynamics [15–17], and light conditions [18–34]. These sensory systems are essential for the survival of organisms, enabling them to avoid harmful environments, locate food sources, find mates, and secure suitable habitats. Complicated information acquired from their sensors is processed through many biochemical and neural pathways. Studying these signal-processing pathways, which have evolved over millions of years, is crucial for understanding organisms' complex searching strategies and adaptive capacities to thrive in diverse ecosystems.

A variety of proteins, ions, complex biological macromolecules, and neurons are involved in processing information and producing appropriate behavioral responses. Information extraction procedures can often be described using the time dynamics of intracellular biochemical reactions that determine how the internal states change under specific inputs or stimuli. One can denote the governing equation of $J$ internal-state variables of interest by $\mathbf{m}(t) = (m_1(t), m_2(t), \cdots, m_J(t))^\top \in \mathbb{R}^J$ and write the evolutionary equation in the general ordinary differential equations (ODEs) form:

$$\frac{d\mathbf{m}}{dt} = \begin{pmatrix} \frac{dm_1(t)}{dt} \\ \frac{dm_2(t)}{dt} \\ \vdots \\ \frac{dm_J(t)}{dt} \end{pmatrix} = \begin{pmatrix} H_1\left(m_1(t), m_2(t), \cdots, m_J(t), s(t)\right) \\ H_2\left(m_1(t), m_2(t), \cdots, m_J(t), s(t)\right) \\ \vdots \\ H_J\left(m_1(t), m_2(t), \cdots, m_J(t), s(t)\right) \end{pmatrix} = \mathbf{H}\left(\mathbf{m}(t), s(t)\right), \tag{1}$$

where $s(t)$ denotes the external stimuli. $\mathbf{H} = \{H_1, H_2, \cdots, H_J\}$ denotes the set of functions that characterize the regulatory relations among $\mathbf{m}(t)$. The particular values of internal variables together with the value of external stimuli determine the behavioral response at the cellular level, which can be quantified by $f$ through an output function $F$:

$$f(t) = F(\mathbf{m}(t), s(t)) \tag{2}$$

Here, the particular forms of $\mathbf{H}$ and $F$ depend on the biological systems and species.

While the general formula above appears straightforward, the specific forms of $\mathbf{H}$ and $F$ may be highly nonlinear. Determining the exact mathematical expressions for $\mathbf{H}$ and $F$ requires both comprehensive knowledge of intracellular biochemical reaction networks and precise quantification of internal state variables. To date, quantitative understanding and the

development of associated mathematical models for signal processing and directed motility have been primarily established for *E. coli* chemotaxis. In contrast, acquiring such measurements of internal reactions in more complex unicellular organisms remains particularly challenging, thereby limiting the applicability of traditional modeling frameworks to these systems.

A natural question arises: Is it possible to discern the governing equations from given stimulus-response data without explicit knowledge of the underlying intracellular biochemical reactions **H** and the behavioral response function *F*? How do we design the external signal and use its characteristic information (values and gradients) to infer the possible underlying structure of intracellular kinetics?

We introduce a novel framework for identifying the governing equations of stimulus-response data, thereby eliminating the need to know the specific forms of Eqs (1) and (2). By reformulating Eqs (1) and (2) using the chain rule, we obtained new equations that are satisfied by the stimulus-response data and their derivatives. The terms in these new equations can be represented by fully connected neural networks, which can be trained using appropriate loss functions. The trained neural network model can predict the cell's temporal response to any type of external stimulus, provided that the stimuli do not change too rapidly. Furthermore, one can infer the number of necessary internal variables for different stimuli.

To validate the proposed approach, we first apply it to numerical experiments using established models of *Escherichia coli* chemotaxis. The results demonstrate that the model accurately captures stimulus-response behavior and effectively recovers key properties of the underlying system. Although the *E. coli* chemotaxis pathway is more complex than models using only one ($J = 1$) or two ($J = 2$) internal variables, when the stimulus does not change too rapidly, most proteins can be considered to be in a quasi-equilibrium state. Given our focus on investigating the quantitative relationship between stimulus inputs and behavioral outputs, previous studies [35–37] have shown that models with one ($J = 1$) or two ($J = 2$) internal variables suffice to reproduce the diverse behavioral patterns observed experimentally. Therefore, this study systematically examines these two cases ($J = 1$ and $J = 2$) as representative scenarios.

Following validation with *E. coli* chemotaxis models, we then applied our method to experimental data of *Euglena gracilis* phototaxis behavior. *E. gracilis*, a eukaryotic microorganism, plays a significant role in ecological systems. These organisms are known for their phototactic responses, which are crucial to ecological phenomena such as algal blooms and diel vertical migration. A wide range of photo-responsive behaviors have been reported for *E. gracilis*, including positive and negative phototaxis, polygonal swimming motion, and localized spinning [38]. When confined to a 2D environment, as in [39], *E. gracilis* exhibits a run-and-tumble pattern and uses phototaxis to redistribute itself based on light intensity. However, the complex photo-receptive systems of *E. gracilis* are difficult to measure directly, and current knowledge about the functional forms of **H** and *F* is limited. It remains uncertain how many internal variables are essential to describe their phototaxis behavior. Our work has achieved two key outcomes for the phototaxis response of *E. gracilis*: First, we derived the governing equation that can predict the response under any given stimulus (though there is a threshold for how rapidly the stimulus can change, due to limitations in the available experimental data). Second, we inferred the potential structure of the underlying intracellular biochemical dynamics.

The insights derived from this study are not accessible via conventional recurrent neural networks (RNNs) [40], neural controlled differential equations (Neural CDEs) [41], or the convolution response kernel method [42,43]. RNNs lack the interpretability required to elucidate the underlying intracellular biochemical dynamics and require equidistant sampling of data points. Furthermore, Neural CDEs encounter difficulties in managing integrals involving

dual internal variables [41]. In contrast, our proposed method circumvents the complexities associated with path integrals and offers a general governing equation. While the convolution response kernel method [42,43] can delineate the relationship between external stimuli and output signals, it is challenging to identify a universal response kernel applicable to diverse external stimuli. More importantly, we are able to achieve good predictive performance with a relatively small training set.

## Methods

### The modeling framework

The main idea of our approach is to use chain rules to eliminate the internal variables. We establish a comprehensive functional relationship between the time derivatives of the stimulus and the tumbling fraction, which is then represented and trained by the neural network approximations.

**Single internal variable model (SIVM).** When there is only a single internal-state variable $m$ in Eq (1), the equation becomes

$$\frac{dm}{dt} = H(m,s). \tag{3}$$

After applying the chain derivative rule to the output function $f(t) = F(m(t), s(t))$ and substituting Eq (3), one has

$$\begin{aligned}
\frac{df}{dt} &= F_m(m,s)\frac{dm}{dt} + F_s(m,s)\frac{ds}{dt} \\
&= F_m(m,s)H(m,s) + F_s(m,s)\frac{ds}{dt},
\end{aligned} \tag{4}$$

where $F_m$ and $F_s$ denote the first-order partial derivative of $F$ in (2) with respect to $m$ and $s$ respectively. Typically, the response $F$ depends monotonically on $m$, as in various *E. coli* chemotaxis models [35,36]. Assume that $F$ in (2) monotonically depends on $m$, then for given $s$ and $f$, one can determine a unique $m$ that satisfies $f = F(m,s)$. This indicates that one can identify a function $M$ such that

$$m = M(f,s). \tag{5}$$

Substituting Eq (5) into Eq (4) yields

$$\frac{df}{dt} = F_m(M(f,s),s)H(M(f,s),s) + F_s(M(f,s),s)\frac{ds}{dt}. \tag{6}$$

The first term $F_m(M,s)H(M,s)$ and the coefficient of the second term $F_s(M,s)$ on the right side of (6) can be expressed as functions that depend only on $f$ and $s$, which we denote as $G_1(f,s)$ and $G_2(f,s)$, respectively. Eq (6) can then be written as:

$$f' = G_1(f,s) + G_2(f,s)s'. \tag{7}$$

The above equation has no internal variable $m$ and depends only on stimulus $s(t)$ and response $f(t)$.

**Dual internal variable model (DIVM).** When two internal states $(m,n)$ are necessary, the dual internal variable reaction can be modeled by the following ODEs:

$$\frac{dm}{dt} = H_1(m,n,s),$$
$$\frac{dn}{dt} = H_2(m,n,s). \tag{8}$$

Similarly, if $F(m,n,s)$ exhibits a monotonic dependence on $m$ and $n$ in the dual internal-state situation, as is valid for the *E. coli* chemotaxis pathway [4,44], one can assume that there exists a function $M$ such that

$$m = M(f,n,s) \tag{9}$$

can be determined. Applying the chain rule to the temporal derivative of $f(t) = F(m(t),n(t),s(t))$ and then using Eqs (8) and (9) yields

$$\begin{aligned}
\frac{df}{dt} &= F_m(m,n,s)\frac{dm}{dt} + F_n(m,n,s)\frac{dn}{dt} + F_s(m,n,s)\frac{ds}{dt} \\
&= F_m(m,n,s)H_1(m,n,s) + F_n(m,n,s)H_2(m,n,s) + F_s(m,n,s)\frac{ds}{dt} \\
&= F_m(M(f,n,s),n,s)H_1(M(f,n,s),n,s) \\
&\quad + F_n(M(f,n,s),n,s)H_2(M(f,n,s),n,s) + F_s(M(f,n,s),n,s)\frac{ds}{dt}.
\end{aligned} \tag{10}$$

The Eq (10) describes a complex relationship among variables $n$, $f$, $s$, and their gradients $f'$ and $s'$, which indicates that one can find a function $N$ that satisfies

$$n = N(f,s,f',s'). \tag{11}$$

Though it is impossible to determine the function $N$ since the internal-state $n$ is not measurable, (11) and (9) indicate that both $m$ and $n$ can be determined by $(f,f',s,s')$.

We introduce the information of the second-order derivative of $f$ and $s$. More precisely, applying the chain rule to the second-order derivative of $f$, one can obtain

$$\begin{aligned}
\frac{d^2f}{dt^2} &= F_m\frac{d^2m}{dt^2} + F_n\frac{d^2n}{dt^2} + F_{mm}\left(\frac{dm}{dt}\right)^2 + F_{nn}\left(\frac{dn}{dt}\right)^2 + F_{ss}\left(\frac{ds}{dt}\right)^2 \\
&\quad + 2\left(F_{mn}\frac{dm}{dt}\frac{dn}{dt} + F_{ms}\frac{dm}{dt}\frac{ds}{dt} + F_{ns}\frac{dn}{dt}\frac{ds}{dt}\right) + F_s\frac{d^2s}{dt^2},
\end{aligned} \tag{12}$$

where

$$\begin{aligned}
\frac{d^2m}{dt^2} &= \frac{\partial H_1}{\partial m}(m,n,s)\frac{dm}{dt} + \frac{\partial H_1}{\partial n}(m,n,s)\frac{dn}{dt} + \frac{\partial H_1}{\partial s}(m,n,s)\frac{ds}{dt} \\
&= (H_1)_m(m,n,s)H_1(m,n,s) + (H_1)_n(m,n,s)H_2(m,n,s) + (H_1)_s(m,n,s)\frac{ds}{dt}, \\
\frac{d^2n}{dt^2} &= \frac{\partial H_2}{\partial m}(m,n,s)\frac{dm}{dt} + \frac{\partial H_2}{\partial n}(m,n,s)\frac{dn}{dt} + \frac{\partial H_2}{\partial s}(m,n,s)\frac{ds}{dt} \\
&= (H_2)_m(m,n,s)H_2(m,n,s) + (H_2)_n(m,n,s)H_2(m,n,s) + (H_2)_s(m,n,s)\frac{ds}{dt}
\end{aligned}$$

Here $F_m$, $F_n$ and $F_s$ denote respectively the first-order partial derivative of $f$ with respect to $m$, $n$ and $s$. $F_{mm}$, $F_{nn}$, $F_{ss}$, $F_{ms}$, $F_{ns}$ and $F_{mn}$ are the second-order partial derivatives. In Eq (12), the first and second-order time derivatives of $m$ and $n$ can all be considered as functions that depend on $m$, $n$, $s$, and $s'$. Thus, the right-hand side of (12) can be expressed as

$$f'' = \bar{G}_3(m, n, s, s') + \bar{G}_4(m, n, s, s')s''. \tag{13}$$

where $\bar{G}_3(m, n, s, s')$ and $\bar{G}_4(m, n, s, s')$ are two undetermined functions. By substituting Eq (9) and Eq (11) into Eq (13), one has

$$f'' = G_3(f, s, f', s') + G_4(f, s, f', s')s''. \tag{14}$$

Eq (14) includes no internal variables $m$, $n$, and depends only on stimulus $s(t)$ and response $f(t)$.

The two models (7) and (14) give general forms that directly connect the signal $s(t)$ and response $f(t)$. However, the particular forms of $G_1(f, s)$, $G_2(f, s)$, $G_3(f, s, f', s')$ and $G_4(f, s, f', s')$ need to be determined from data and we use deep neural networks (DNN) to represent them in the subsequent part.

*Remark.* The monotonicity of the response function $F$ with respect to internal variables is essential for inverting the function. While this assumption holds for E. coli chemotaxis, its generalizability to other microorganisms can be reasoned as follows. The central idea of our reformulation is this: the quantities $m$ and $n$ are unknowns that could potentially represent the concentration of a specific protein or a function derived from it. This approach allows $m$ and $n$ to be defined flexibly, such that they do not inherently need to correspond to the concentration of a specific protein. We assume one key protein concentration is $c$, and define $m = \phi(c)$. By making no assumptions about the specific form of $\phi$, we can always posit the existence of a function $\phi$ such that $F$ is monotonic with respect to $\phi(c)$, even if $F$ is non-monotonic with respect to $c$ itself.

## Chemotaxis models in *E. coli*

A well-studied example of signal processing and behavioral response is the run-and-tumble *E. coli* chemotaxis system. *E. coli* can move toward favorable environments through alternating run-and-tumble patterns by rotating flagella, governed by an internal chemosensory system, which can be described in the form of (1) and (2):

- **The intracellular mechanism of the *E. coli* chemosensory system.** The internal biochemical reactions in *E. coli* is of the form (1) and can be highly complex. The forms and numbers of ODEs differ significantly in different models. One type of *E. coli* model considers the molecular details including receptor clusters and a coupled biochemical reaction network [45–53]. The other category considers only the system-level properties of the signaling pathway, which employs a receptor clustering-based framework and takes the signaling complex as the basic unit [35–37,54–58]. Some complex models can finely describe the dynamics of internal variables with up to 158 ODEs [58]. The signalling pathways that primarily govern cell motility can often be described by 1 to 2 ODEs [35,37].
- **The behavioral response.** Flagella of each *E. coli* rotate either counter-clockwise (CCW) or clockwise (CW). When most flagella rotate CCW, the bacterium runs straight; when

most flagella rotate CW, the cell stops and changes its movement direction. The concentration of the response regulator protein affects the rotational direction of the flagella, thus the probability that the bacterium is in the running or tumbling phase. The behavioral response of *E. coli* cells can be described by a tumbling fraction (the probability that the cells are in the tumbling state) or tumbling frequency, which can be modeled by the equation of the form (2). These two quantities are also modeled by functions of various forms. For example, the tumbling fractions take the following forms: $f = F(m_1, s)$ in [35], $f = F(m_1)$ in [44], or $f = F(m_1, m_2, s)$ in [4]. The specific forms of $F$ vary in the literature, depending on the experimental setup and bacterial strains.

In the past decades, great progress has been made in understanding the mechanism of the chemosensory system and behavioral response, leading to various mathematical models. Here, we present an exposition of three representative models in the literature: one reduced model with $J = 1$ and two comprehensive dynamical models with $J = 2$.

The well-studied *E. coli* chemosensory system mainly adapts to the relatively slow receptor methylation and demethylation processes catalyzed by two enzymes CheR and CheB, which can be described by one ODE. The slow dynamics of receptor methylation level, denoted as $m$, and tumbling fraction, denoted as $f$, can be generally modeled by the following single internal-state model I [36,44]:

$$\frac{dm}{dt} = H_{I,1}(m, s),$$
$$f = F_I(m, s). \tag{Model I}$$

Model I for $m$ is typically coupled with equations for the downstream phosphorylation and dephosphorylation reactions of CheA, CheY, and CheB. Since these reactions occur much faster than the methylation and demethylation reactions, one can use quasi-steady state (QSS) approximations to get the simplified Model I. The specific forms and parameter values of $H_{I,1}$ and $F_I$ can be found in Appendix A of S1 Text, and the validation of the monotonic dependence of $F_I$ on $m$ can be found in Fig E(A) in S1 Text.

The second model takes into account the excitation of the signal transduction pathway, in which CheY-P plays an important role. The changes in the methylation level ($m$) [35] and the CheY-P concentration ($Y_p$) [53] can be modeled by the evolution equation of the following form with $f$ determined only by $Y_p$ [44]:

$$\frac{dm}{dt} = H_{II,1}(m, s),$$
$$\frac{dY_p}{dt} = H_{II,2}(m, Y_p, s),$$
$$f = F_{II}(Y_p). \tag{Model II}$$

The temporal scales of $m$ and $Y_p$ differ significantly. The methylation level $m$ changes on a much slower time scale (ranging from seconds to minutes) compared to the CheY-P concentration, which evolves on a timescale of less than 1 s [53]. The specific forms and parameter values of $H_{II,1}$, $H_{II,2}$ and $F_{II}$ can be found in Appendix A of S1 Text, and the validation of the monotonic dependence of $F_{II}$ on $Y_p$ can be found in Fig E(B) in S1 Text.

A third model with dual internal states that do not have temporal scale differences can be found in [4]. Based on the findings that changes in CheY-P levels during adaptation lead to changes in FliM proteins, the dynamics of FliM proteins FliM and receptor methylation level

$m$ in Model III are:

$$\frac{dm}{dt} = H_{\text{III},1}(m, s),$$

$$\frac{d}{dt}\text{FliM} = H_{\text{III},2}(m, \text{FliM}, s), \qquad \text{(Model III)}$$

$$f = F_{\text{III}}(m, \text{FliM}, s).$$

Here $F_{\text{III}}$ depends monotonically on $m$ and FliM (see Fig E(C-D) in S1 Text). The temporal scales of $m$ and FliM in Model III are similar.

These three models have connections, yet are distinct. As shown in Fig F in S1 Text, their different responses to a sudden change in $s(t)$ are displayed. Model I is based on detailed biochemistry reactions, where the CheY-P level $Y_p$ is determined by $m$ and $s$ through QSS approximation. When $s(t)$ changes fast, in a short time scale, the QSS approximation no longer holds, necessitating the use of $H_{\text{II},2}(m, Y_p, s)$ in Model II to describe the dynamics of $Y_p$. Model I and II can explain the "perfect adaptation" [59], which indicates that when cells are exposed to sudden environmental changes, they respond by temporarily changing their tumbling frequency, then gradually returning to their pre-stimulus tumbling frequency over time. On the other hand, Model III emphasizes the "overshoot" feature during the adapta-tion process, quantifying the degree of excessive response before the return to pre-stimulus behavior. The particular forms of $H_{\text{I},1}(m, s)$ in [36], $H_{\text{II},1}(m, s)$ in [35], and $H_{\text{III},1}(m, s)$ in [4] are different in different works, but the last two are linear approximation of the first one. More details of Models I-III are included in Appendix A in S1 Text. Given the diversity of *E. coli* models, identifying the appropriate model to describe a new experimental dataset presents a challenge.

Due to the complexity of model selection and the difficulty of determining the number of internal states, it becomes essential to establish governing equations that give the relationship between stimuli and response signals without relying on information about internal reactions. Although the internal chemical reactions within cells may be highly complex, the key reac-tions affecting their movement behavior can be described using only 1-2 ODEs, as in the *E. coli* chemotaxis pathway.

## Photo-response movement in algae

Photosynthetic microorganisms, such as microalgae, exhibit phototaxis—directional move-ment in response to light stimuli [22,25,26]. For example, *E. gracilis* adjusts its flagellar beat-ing patterns in response to perceived light intensity, enabling navigation toward or away from light sources [23–25,29,33,60–63]. In microalgae, photoreceptors convert optical signals into electrical signals via electron/ion interactions, which are then transmitted to locomotor organelles like flagella.

- **The possible intracellular mechanism of the *E. gracilis* photoreception system** The photoreceptor has been revealed to be photoactivated adenylyl cyclase, which produces cyclic adenosine monophosphate (cAMP) upon blue light exposure. Intracellular cAMP levels rise sharply within 1 s of illumination before rapidly returning to baseline. This transient cAMP signal may activate a protein kinase, which could phosphorylate one or several flagellar proteins to alter beating patterns, ultimately regulating phototactic behavior of *E. gracilis* [64]. However, no model is available to quantify the intracellular signaling pathway.
- **The behavioral response.** When confined to a 2D environment, as reported in [39], *E. gracilis* exhibits a run-and-tumble movement pattern. Tumbling is defined as a

decrease in translational velocity or a concurrent increase in angular velocity [6,39]. The behavioral response can be quantified by the tumbling fraction, which displays complex patterns in response to changes in light intensity.Details of the experimental setup and measurement method are in Appendix B in S1 Text.

Adaptive photo-responses have been observed in other algae species, such as Chlamydomonas reinhardtii [34,65] and Volvox carteri [21]. These species exhibit a similar response pattern: an initial rapid increase in the signal, followed by a slower recovery to the resting state. Models of the form given by Eqs (1) and (2) with $J = 2$ have been proposed in the literature to describe the photo-responses of Chlamydomonas reinhardtii [34] and Volvox carteri [21]. However, the light-response mechanisms of *E. gracilis* remain poorly understood. One can hypothesize that its photoresponse model shares a similar underlying structure.

### Implementation

Experimental data typically contain noise, which can be modeled by adding a white noise $\epsilon$ of Gaussian type with zero mean and standard deviation $\sigma_\epsilon$. The measured data points $y_i$ are given by

$$y_i = f_i + \epsilon_i, \ \epsilon_i \sim \mathcal{N}\left(0, \sigma_\epsilon^2\right), \ i = 1, 2, \cdots, N^{\text{data}}. \tag{15}$$

Noisy data cannot be accurately differentiated and requires prior smoothing. One can smooth the data sequence using a spline method to get the reference solutions. We apply the cubic smoothing spline method to noisy data points, and the reference is the minimizer of

$$\min_{\tilde{f}} \quad p \sum_{i=1}^{N^{\text{data}}} |y_i - \tilde{f}(t_i)|^2 + (1 - p) \int |D^2 \tilde{f}(t)|^2 dt.$$

Here, the first term measures the error and the second term controls the roughness. $D^2\tilde{f}$ denotes the second order derivative of the function $\tilde{f}$. One can adjust the smoothing parameter $p$ to regulate the smoothness of the reference solution. For the discontinuous responses, such as those under step stimuli, the smoothing method will be used to smooth the noisy data for each segment of constant stimulation, rather than smoothing the entire data sequence. We use the "csaps" function in MATLAB to achieve the process. Following the analysis provided in [66], the discrepancy between the smoothed value $\tilde{f}_i$ and the theoretical value $f_i$ is negligible. In our subsequent part, we adopt the smoothed data as the reference and denote points in this data set by $f_i$.

Based on Eqs (7) and (14), deep neural networks $G_{\theta_1}^{NN}(f, s)$, $G_{\theta_2}^{NN}(f, s)$, $G_{\theta_3}^{NN}(f, s, f', s')$ and $G_{\theta_4}^{NN}(f, s, f', s')$ can be constructed and trained to approximate the corresponding functions $G_1(f, s)$, $G_2(f, s)$, $G_3(f, s, f', s')$ and $G_4(f, s, f', s')$.

In the SIVM, we train the function based on the function values of the discrete-time points $t_i$ ($i = 1, 2, \cdots, N^{\text{train}}$). The stimulus and responses at these discrete time points are $s_i$ and $f_i, f'_i$ is the approximation of $f'(t_i)$. At all discrete-time $t_i$, $G_{\theta_1}^{NN}(f_i, s_i)$ and $G_{\theta_2}^{NN}(f_i, s_i)$ should satisfy Eq (7). Therefore, the SIVM loss is defined as

$$\mathcal{L}_{\text{SIVM}} = \frac{1}{N^{\text{train}}} \sum_{i=1}^{N^{\text{train}}} \left(f'_i - G_{\theta_1}^{NN}(f_i, s_i) - G_{\theta_2}^{NN}(f_i, s_i) s'_i\right)^2. \tag{16}$$

Similarly, given $(f_i, s_i, f'_i, s'_i)$, the DIVM loss based on Eq (14) is

$$\mathcal{L}_{\text{DIVM}} = \frac{1}{N^{\text{train}}} \sum_{i=1}^{N^{\text{train}}} \left( f''_i - G^{NN}_{\theta_3}(f_i, s_i, f'_i, s'_i) - G^{NN}_{\theta_4}(f_i, s_i, f'_i, s'_i) s''_i \right)^2. \tag{17}$$

Denote the time step by $\Delta t$, one can approximate $f'_i, s'_i$ in (16) and (17) by

$$f'_i = \frac{f_i - f_{i-1}}{\Delta t}, \ s'_i = \frac{s_{i+1} - s_{i-1}}{2\Delta t}; \tag{18}$$

$f''_i$ and $s''_i$ by

$$f''_i = \frac{f_{i+1} - 2f_i + f_{i-1}}{(\Delta t)^2}, \ s''_i = \frac{s_{i+1} - 2s_i + s_{i-1}}{(\Delta t)^2}. \tag{19}$$

It should be emphasized that, when constructing the losses, the idea is similar to that of physics-informed neural networks (PINN) [67–69], where physical laws expressed by differential equations and the chain rule are employed. The SIVM/DIVM loss resembles the PDE loss in PINN, but the $f'$ and $f''$ are not derived from the automatic differentiation, and the time variable $t$ does not appear explicitly in the neural network. Moreover, the time variable is only used to provide the approximations in (18) and (19). As far as one can get the time derivatives, there is no need to use equally spaced discrete time points, which differs from RNNs [40]. Therefore, during training, one can use data from different time series and cut the series freely.

The data are divided into three parts: 70% for training, 10-15% for validation, and the remainder for testing. Owing to the inherent noise in the experimental data of $f$, even after smoothing, attempting to approximate $f'$ can lead to the amplification of error. This may lead to overfitting during the training, thereby affecting the prediction accuracy of $f$. To avoid overfitting, during training the functions $G^{NN}_{\theta_1}(f, s)$, $G^{NN}_{\theta_2}(f, s)$, $G^{NN}_{\theta_3}(f, s, f', s')$ and $G^{NN}_{\theta_4}(f, s, f', s')$ by the training data, one has to test the results on the validation set. The obtained functions are validated by the function values of $f$ in the validation set. More precisely, given the initial values of $f$ in the validation set, such that

$$\hat{f}_{\text{initial}} = \begin{cases} \hat{f}_1 = f(t_1), & \text{for SIVM,} \\ \hat{f}_1 = f(t_1), \hat{f}_2 = f(t_2), & \text{for DIVM,} \end{cases}$$

one can iteratively obtain $\hat{f}_{i+1}$ by the following equations:

$$\begin{aligned} \frac{\hat{f}_{i+1} - \hat{f}_i}{\Delta t} &= G^{NN}_{\theta_1}(\hat{f}_i, s_i) + G^{NN}_{\theta_2}(\hat{f}_i, s_i) s'_i, \quad \text{for SIVM,} \\ \frac{\hat{f}_{i+1} - 2\hat{f}_i + \hat{f}_{i-1}}{(\Delta t)^2} &= G^{NN}_{\theta_3}(\hat{f}_i, s_i, \hat{f}'_i, s'_i) + G^{NN}_{\theta_4}(\hat{f}_i, s_i, \hat{f}'_i, s'_i) s''_i, \quad \text{for DIVM.} \end{aligned} \tag{20}$$

Then the long-term responses can be iteratively given by

$$\hat{f}_{i+1} = \begin{cases} \hat{f}_i + \Delta t \left( G^{NN}_{\theta_1}(\hat{f}_i, s_i) + G^{NN}_{\theta_2}(\hat{f}_i, s_i) s'_i \right), \ i \geqslant 2 \text{ for SIVM,} \\ -\hat{f}_{i-1} + 2\hat{f}_i + (\Delta t)^2 \left( G^{NN}_{\theta_3}(\hat{f}_i, s_i, \hat{f}'_i, s'_i) + G^{NN}_{\theta_4}(\hat{f}_i, s_i, \hat{f}'_i, s'_i) s''_i \right), \ i \geqslant 3 \text{ for DIVM.} \end{cases} \tag{21}$$

The mean squared errors are employed to evaluate the accuracy of the validation set, whose definition is

$$E_{\text{vali}} = \frac{1}{N^{\text{vali}}} \sum_{i=1}^{N^{\text{vali}}} \left( \hat{f}_i - f_i \right)^2 . \tag{22}$$

We terminate the training process until $E_{\text{vali}}$ stops decreasing significantly. An illustrative schematic is shown in Fig 1.

Upon completion of the training process, we use the same methodology in Eq (21) to predict the function values of $\hat{f}$ on the test data set. The prediction accuracy is quantified using the relative mean squared error:

$$E_{\text{test}} = \frac{1}{N^{\text{test}}} \sum_{i=1}^{N^{\text{test}}} \left( \frac{\hat{f}_i - f_i}{f_i} \right)^2 . \tag{23}$$

## Results

Parameters of DNN are trained upon the losses defined by Eq (16) or Eq (17). The DNN architecture comprises feed-forward neural networks with 4 hidden layers and 20 neurons

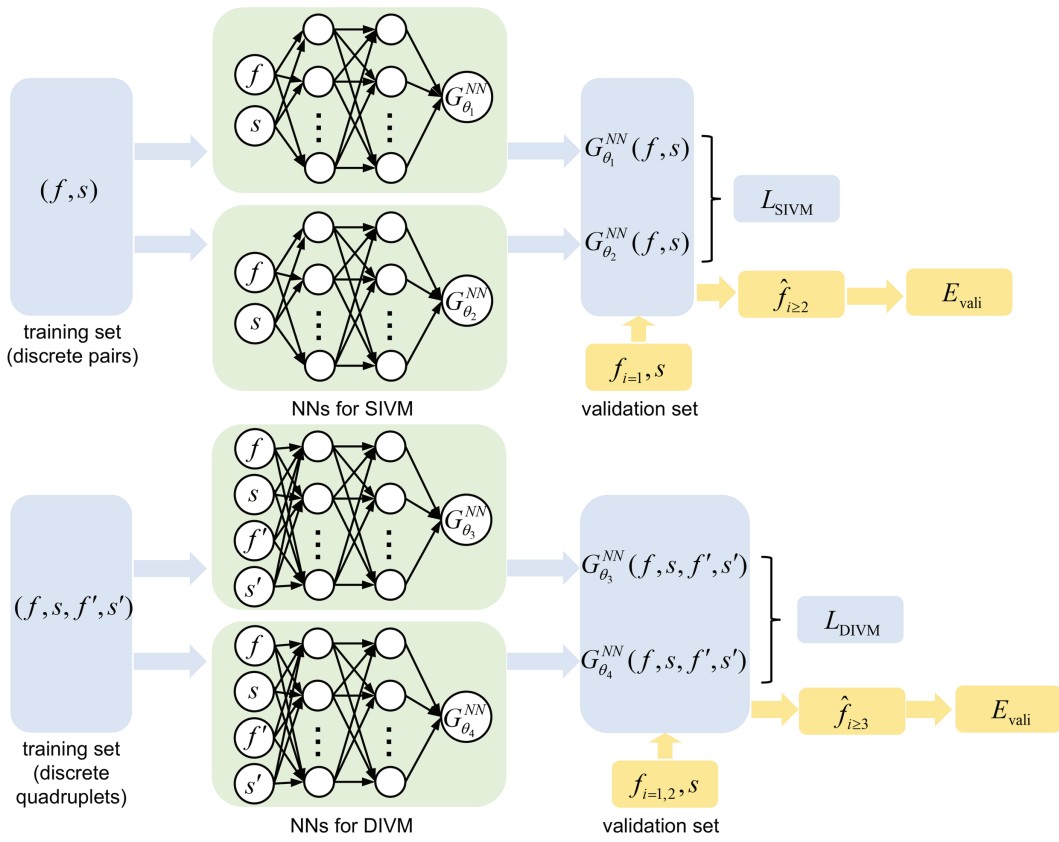

**Fig 1. Schematic of SIVM and DIVM.** The input dataset for SIVM and DIVM are respectively discrete pairs $(f_i, s_i)$ and quadruplets $(f_i, s_i, f_i', s_i')$ $(i = 1, 2, \cdots, N^{\text{train}})$. The neural networks $G_{\theta_1}^{NN}(f, s)$ and $G_{\theta_2}^{NN}(f, s)$ are trained through SIVM loss $\mathcal{L}_{\text{SIVM}}$, while $G_{\theta_3}^{NN}(f, s, f', s')$ and $G_{\theta_4}^{NN}(f, s, f', s')$ are constrained by DIVM loss $\mathcal{L}_{\text{DIVM}}$. During the training of neural networks, we assess the performance of the validation set through $E_{\text{vali}}$ between $\hat{f}_i$ and $f_i$ to avoid overfitting.

in each hidden layer. We use the tanh activation function and choose the Glorot uniform method for the weight initialization. For the training, we use an Adam optimizer with default hyperparameters and a learning rate of $10^{-3}$. $E_{\text{vali}}$ in (22) is outputted every 50 or 100 training epochs during the training process.

## Performance on the *E. coli* tumbling fraction

We first study the performance of our algorithms for the *E. coli* models when adding noise to the data in the training set. The training data are generated by using the following two different types of $s(t)$:

- **Piecewise constant (PWC) functions.** $s(t)$ across the temporal domain $[T_{\min}, T_{\max}]$ is a piecewise constant function:

$$s(t) = \sum_{k=1}^{K^t} C_k(t)$$

where the time domain is randomly divided into $K^t$ intervals

$$T_{\min} = T_1 < T_2 < \cdots < T_k < T_{k+1} < \cdots < T_{K^t+1} = T_{\max},$$

and

$$C_k(t) = \begin{cases} c_k, & \text{if} \quad t \in [T_k, T_{k+1}) \\ 0, & \text{otherwise} \end{cases}$$

$c_k$ is a constant randomly selected between $[s_{\min}, s_{\max}]$.
- **Linear combination of cosine (LCC) functions.** The linear combination of cosine functions is defined as:

$$s(t) = \frac{1}{P} \sum_{p=1}^{P} a_p \left( 1 - \cos\left( \frac{2\pi t}{b_p} \right) \right),$$

where $a_p$ and $b_p$ are randomly chosen from a uniform distribution of the intervals $[a_{\min}, a_{\max}]$ and $[b_{\min}, b_{\max}]$, respectively.

An illustration of the two types of stimulus is presented in Fig G(A, C) in S1 Text.

We generate tumbling responses $f_i$ using a forward Euler discretization. To represent experimental noise, we introduce Gaussian noise $\epsilon_i$ with a mean of zero and a standard deviation $\sigma_\epsilon$ through a given signal-noise ratio (SNR) which can be calculated by

$$\sigma_\epsilon = \sqrt{P_{\text{noise}}} = \sqrt{\frac{P_{\text{signal}}}{10^{\text{SNR}/10}}}.$$

For the $j$-th numerical solution with $N_j^{\text{data}}$ data points, the power $P_{\text{signal}}$ of the original signal can be achieved by calculating the mean square value of the original signal: $P_{\text{signal}} = 1/N_j^{\text{data}} \sum_{i=1}^{N_j^{\text{data}}} f_i^2$. We set SNR to a range of 20 to 22, which aligns with the measured *E. gracilis* experimental data, which we will discuss later.

Here, we use stimulus-response data obtained by simulating Model I-III and then adding some noise. The particular choices of $s_{\min}$, $s_{\max}$, $a_{\min}$, $a_{\max}$, $b_{\min}$, and $b_{\max}$ relate to the time scale of the behavioral response. Due to the related experimental phenomena, different *E. coli*

models have different response time scales, which play a role in the design of the data sets. That is, the stimuli in the test set can have different forms, but their magnitudes and rates of change have to be in a similar range to those in the training set. Detailed settings for each model's training, validation, and test sets are in Appendix D in S1 Text.

It should be noted that, during training and testing, we only use data with $s'(t)$ and $s''(t)$ that are finite. More precisely, we exclude those points near the jump discontinuities of $s(t)$ and $s'(t)$. We train the NNs for each model using the SIVM and DIVM losses separately. After training, we determine that the NNs are not overfitted by monitoring the error decrease Eq (22) on the validation set.

We then enrich the testing set with smooth stimuli that do not belong to the two types of basic stimuli. Due to the fold-change detection of *E. coli* cells [70], in [3,48,53], exponential sine wave signal $s(t) = \exp(a_0 cos(2\pi t/b_0))$ is used to investigate the cells' response. We test our scheme's performance using a generalization of the exponential sine signal and consider the following more complex stimuli:

- **Exponential functions with exponents being a linear combination of cosines (ELCC):**

$$s(t) = exp\left(\sum_{q=1}^{Q} a_q\big(1 - cos(2\pi t/b_q)\big) - c_q\right).$$

  where $a_q$, $b_q$ and $c_q$ are chosen randomly in a interval.

For testing, we compare the predicted dynamics inferred by both algorithms $\hat{f}_i$ based on Eq (21), with the reference solution. The results are shown in Fig 2 and very good agreement can be observed. For ELCC signals used in Fig 2(A)–2(C), the range of $a_q$, $b_q$ is consistent with that used in the LCC functions. $c_q$ is employed to adjust the magnitude of $s(t)$ to satisfy $s(t) \in (s_{\min}, s_{\max})$ with $s_{\min}$ being the maximum (minimum) of the stimuli in the training set.

Model I is a model with a single internal variable. As shown in Fig 2(A), the predicted values obtained from both algorithms are very close to the reference values, indicating that both SIVM and DIVM can effectively handle the situation with a single internal variable. Model II has two internal variables, and there is a significant reaction time scale difference between the two variables. The results in Fig 2(B) demonstrate that DIVM performs exceptionally well in predicting the test data. At the same time, SIVM, although slightly less accurate, still provides relatively accurate predicted values. Unlike Model II, Model III considers two internal variables with no significant difference in reaction speeds. Fig 2(C) shows that the performance of SIVM is significantly inferior to that of DIVM. These results suggest that DIVM can accurately give good long-term predictions for signal patterns that do not appear in the training set, for single and dual internal variable models. On the other hand, SIVM works for data generated with a single internal variable model and sometimes dual internal variables as well.

The performance of SIVM and DIVM for different datasets offers a way to infer the required number of internal variables to model dynamics that produce specific response signals. By comparing the performance of SIVM and DIVM on datasets from Model I and Model III, we can deduce that if both algorithms perform well, the dataset can be modeled by the response of a single internal variable. Conversely, if only DIVM performs well, it suggests that the internal biochemical reactions corresponding to the dataset need to be described by two ODEs, as shown in Fig 2(C).

We further investigate our algorithms' robustness to complex stimuli with stochastic fluctuations. Specifically, the noisy stimuli are given by $s_i = s_i^{\text{origin}} + \epsilon_i$ where $s^{\text{origin}}$ represent noiseless PWL, LCC, and ELCC signals. We generate tumbling responses $f_i$ by simulating Model

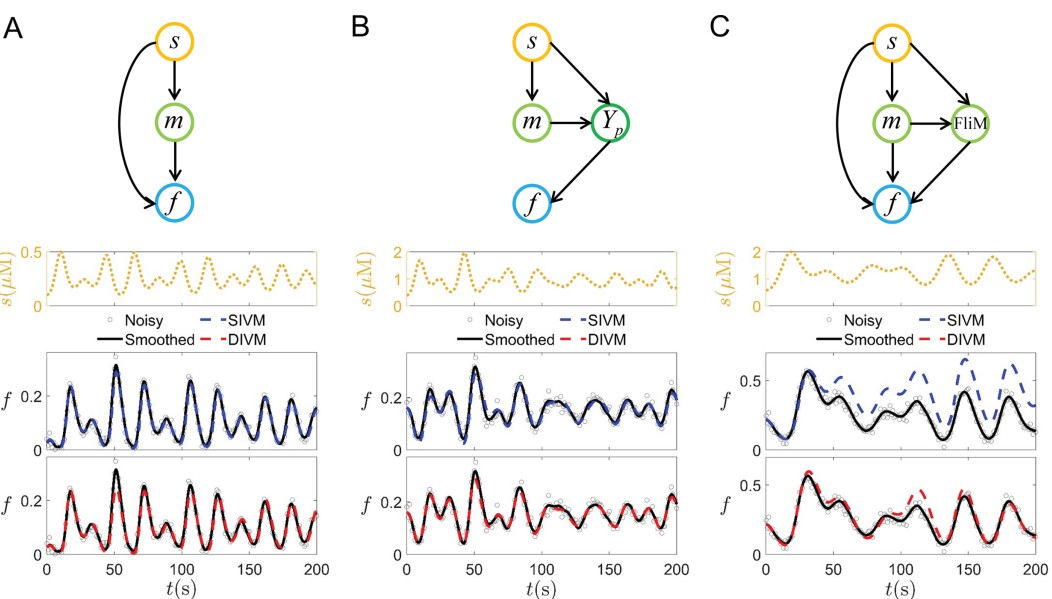

**Fig 2. Performance of SIVM and DIVM on *E. coli* models I-III under ELCC signals.** (A) Model I; (B) Model II; (C) Model III. Top row: the simplified topological relationships among external stimuli, internal variables, and tumbling fractions in the three different *E. coli* models. $x \to y$ denotes that $x$ influences $y$, which can be both excitation or inhibition. Second row: The outside signal $s(t)$ of the form ELCC. The third row: the predictive performance of SIVM. The fourth row: the predictive performance of DIVM. Here, the response data (gray circles) are obtained by adding Gaussian noise to the numerical solutions of the corresponding models with SNR following the uniform distribution $U(20,22)$. These noisy data, after smoothing, serve as the reference values (black solid line).

I-III based on these noisy stimuli (as shown in Fig B in S1 Text). The cubic smoothing spline method is applied to smooth the data sequences $s(t)$ and $f(t)$, from which the derivatives $f_i'$, $s_i'$, $f_i''$, and $s_i''$ are approximated. During both training and testing, we utilize noisy stimulus-response data $(f_i, s_i)$. The performance of both algorithms in predicting response data using noisy stimuli is similar to that using smooth stimuli, indicating that our algorithms are robust to complex stimuli with stochastic fluctuations. Although noise levels can degrade model performance (see Table E in S1 Text), our algorithms guarantee stable long-term predictions of $f(t)$ without significant noise-induced fluctuations.

## Performance on the measured *E. gracilis* tumbling fraction

Next, we extend the application of our algorithms to experimental data of *Euglena gracilis* cells. *E. gracilis* move with similar run-and-tumble patterns with *E. coli*, as illustrated in Fig 3(A), while the biological pathways for their photo-responsive movement are not yet fully understood. During "runs", the cell moves in a near-ballistic manner, whereas in "tumbles", it spins before selecting a new direction for the subsequent "run". Then, the fraction of tumbling particles during every second can be quantified (shown in Fig 3(B)–3(E)) under varied light-intensity stimuli $s(t)$ ($s(t)$ is the light intensity whose unit is W/m$^2$).

In the experiment, we have measured the corresponding tumbling fraction of *E. gracilis* for different $s(t)$. Three different types of signal $s(t)$ are used, PWC, LCC and

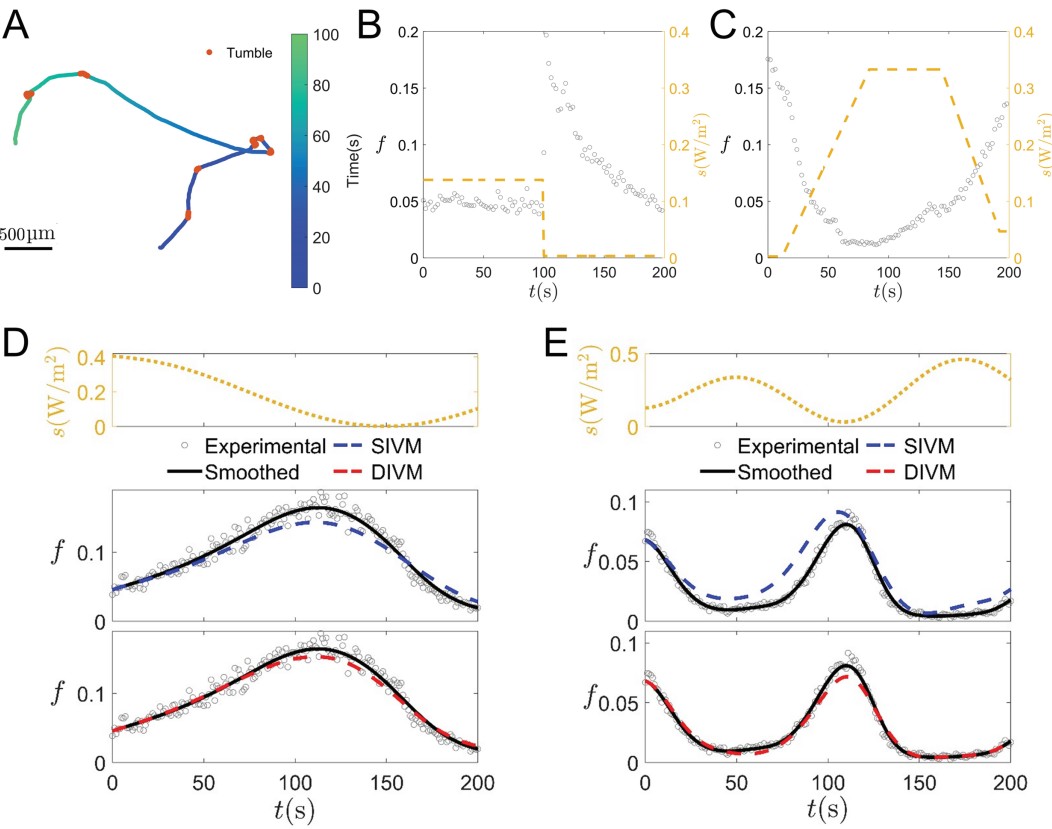

**Fig 3. Performance of SIVM and DIVM on *E. gracilis* experimental data.** (A) Typical cell trajectory with uniform light intensity, where the red dots highlight tumble events. (B-C) Measured tumbling fraction with PWC and PWCL stimuli. (D-E) Performance of SIVM and DIVM for different LCC stimuli (yellow dotted line). Gray circles are the experimental data. (D) is for $s'_{max} = 4.18 \times 10^{-3}$ W/(m²·s) and (E) is for $s'_{max} = 10.441 \times 10^{-3}$ W/(m²·s). Top row: the LCC stimuli; middle row: predictive performance of SIVM; bottom row: predictive performance of DIVM.

- **Piecewise constant and linear (PWCL) functions.** $s(t)$ is a combination of piecewise constant and linear functions:

$$s(t) = \sum_{k=1}^{K^t} C_{2k-1}(t) + \sum_{k=1}^{K^t} L_{2k}(t),$$

where the time domain is randomly divided into $2K^t$ intervals

$$T_{min} = T_1 < T_2 < \cdots < T_k < \cdots < T_{2K^t+1} = T_{max}.$$

For $k = 1, \cdots, K^t$, the linear functions are defined as

$$L_{2k}(t) = \begin{cases} c_{2k-1} + K_k^*(t - T_{2k}), & \text{if} \quad t \in (T_{2k}, T_{2k+1}) \\ 0, & \text{otherwise} \end{cases}$$

where

$$K_k^* = \frac{c_{2k+1} - c_{2k-1}}{T_{2k+1} - T_{2k}}.$$

The values of $K_k^*$ encompass the gradient information of $s(t)$. Therefore, the ranges of $K_k^*$ in the training and test sets affect the precision. An illustration of this stimulus type is presented in Fig G(B) in S1 Text.

Given that the size of *E. gracilis* is significantly larger than that of *E. coli*, its movement is considerably slower. Consequently, it is crucial to control the variation rate of $s(t)$, which can be quantified by $s'(t)$, in the designed light signals. Two different datasets are used to test the performance of our proposed method, one with slow varying $s(t)$ while the other includes both slow and relatively faster varying $s(t)$. We use the absolute maximum values of the stimuli, denoted as $s'_{max} = \max_{t\in[0,T]} |s'(t)|$, to characterize these two different datasets. The stimulus types and $s'_{max}$ values used in the two training sets are shown in Table 1. The smoothing spline method is applied to the raw data to obtain the smoothed reference values. Subsequently, SIVM and DIVM are employed to identify the governing equation of the run-and-tumble dynamics through the data set composed of external stimuli $s$ and responses $f$. The details for each experiment's training and test sets are shown in Appendix D in S1 Text.

The predicted tumbling fractions under LCC signals are shown in Fig 3(D-E). Using slowly varying $s(t)$, the prediction errors of both SIVM and DIVM are below 3%, with DIVM outperforming SIVM. This suggests that when $s'_{max} = \max_{t\in[0,T]} |s'(t)| < 4.183\times10^{-3}$ W/(m$^2$·s), the governing equation can be obtained using SIVM, and a single ODE can describe the internal dynamics. However, when $s(t)$ changes faster, the prediction error of SIVM is much higher than the acceptable 10% threshold, and SIVM fails to provide an accurate governing equation. After retraining with DIVM, the accuracy of prediction is significantly improved, as shown in Table 1, indicating that the internal dynamics of fast-changing environments with $s'_{max} = \max_{t\in[0,T]} |s'(t)| < 10.441\times10^{-3}$ W/(m$^2$·s) require two ODEs for an accurate description.

Comparisons between the two datasets reveal that *E. gracilis* exhibits distinct photo-response patterns depending on the rate of change of external signals. In slowly changing environments, a single internal variable suffices to model the response, whereas in rapidly changing environments, two internal variables with clearly distinct reaction timescales are required. According to the biological signal transduction mechanisms of *E. gracilis* explored in [64], one can hypothesize that the fast internal variable may correlate with the rapid cAMP-mediated phosphorylation of flagellar proteins. The underlying mechanism for slow-scale responses remains unclear.

The good performance of SIVM in slowly varying environments is similar to the case of Model II in the test for *E. coli* models. Therefore, in the next subsection, we will generate a series of data sets with different $s'_{max}$ based on Model II to further investigate the conditions under which SIVM or DIVM can be applied.

## Relation between SIVM and DIVM and criteria for their selection

Notably, Model II includes two internal variables, but SIVM can predict the behavioral response. Further tests indicate that when $s'(t)$ remains below a certain threshold, SIVM can

**Table 1. Test errors of *E. gracilis* experiments.**

| Stimulus | $s'_{max}$(W/(m$^2$·s)) | Training set | Algorithm | $E_{test}$ (%) |
|---|---|---|---|---|
| Slow | $4.183\times10^{-3}$ | PWC, LCC | SIVM | 2.24% |
|  |  |  | DIVM | 0.35% |
| Fast | $10.441\times10^{-3}$ | PWCL, LCC | SIVM | 59.63% |
|  |  |  | DIVM | 1.64% |

yield good predictions, but when $s'(t)$ increases, the predictive efficacy of SIVM degrades. This can be understood by observing that the reaction rates of the two internal variables are significantly different in Model II. When $s(t)$ changes slowly, one of the two internal variables is always in a quasi-equilibrium state. Hence, the model degenerates to a one-internal-variable model, and SIVM can predict the response. This observation raises a natural question: Is it possible to identify the threshold $S_g$, when $\max_t s'(t) < S_g$, SIVM can provide a good prediction, while when $s'(t) > S_g$, one has to use DIVM. In this subsection, we will explore this issue.

The threshold $S_g$ can be derived using analytical approaches. Without loss of generality, we assume the variable $n$ in Eq (8) is the fast variable and $n_{ss}$ is the quasi-equilibrium state that satisfies $H_2(m, n_{ss}, s) = 0$. The applicability of SIVM implies that $n(t) = n_{ss}$ and one internal variable $m$ suffices. Conversely, if $n(t)$ deviates from $n_{ss}$, we consider a small perturbation $\delta n(t)$ around $n_{ss}$ such that

$$n(t) = n_{ss}(m(t), s(t)) + \delta n(t).$$

When the deviation $\delta n(t)$ satisfies $|\delta n(t)| < \Delta_n$ (where $\Delta_n$ is a specified small threshold), the theoretical expression for the threshold $S_g$ is given by:

$$S_g = \Delta_n \max_{s,m} \left| \frac{\frac{\partial H_2}{\partial n}(n_{ss})}{\frac{\partial n_{ss}}{\partial s}} \right|.$$

Details of the derivation are provided in Appendix C of S1 Text. For Model II, the threshold $S_g$ can be computed by setting, for example, $\Delta_n = 0.1$, $s \in [0,2]$ μM and corresponding $m \in [1, 1.07]$ near the reference methylation level, reducing the above expression to:

$$S_g = \frac{\Delta_n}{\tau_Z} \cdot \max_{m \in [1,1.07], s \in [0,2]} \left| \frac{(A(m,s))^{-2}(1 + s/K_I)}{N_r k_a \tau_Z \exp(N_r F_A(m,s))\left(1/K_I - \frac{1+s/K_I}{K_A+s}\right)} \right| \approx 0.59 \text{ μM/s},$$

where $\tau_Z = 0.5$ s represents the characteristic timescale of $Y_p$ and details of all other parameters can be found in Appendix A of S1 Text. Thus, the threshold $S_g$ is closely tied to the biological parameters of the underlying model, provided that the explicit formulations are known.

Alternatively, the threshold can be estimated numerically by testing the algorithm's performance across varying rates of change in $s(t)$ (denoted as $s'(t)$). We conduct a systematic training and test process with datasets that vary in $s'(t)$. More precisely, for given external signals $s(t)$ belonging to LCC or ELCC, if the total time of a continuous segment is $T$, $s(t)$ is progressively compressed by $s\left(t\frac{T_1}{T}\right)$ with $T_1 < T$, as shown in Fig 4(A). As $T_1$ decreases, $s'_{max}$ increases. In Fig 4(B), one can observe that SIVM fails to give good predictions effectively, based on the data produced by Model II when $|s'(t)|$ becomes large.

We correlate SIVM and DIVM test errors with the absolute maximum values of $s'_{max} = \max_{t \in [0,T]} |s'(t)|$ of different data sets, as shown in Fig 4(B). SIVM test errors increase as $s'_{max}$ increases, whereas DIVM exhibits low test errors across all data sets. When $s'_{max}$ exceeds a certain threshold, the testing error increases significantly. Based on this, we numerically establish a threshold for $s'_{max}$; exceeding this threshold suggests that SIVM is no longer suitable. By requiring that the relative errors in (23) of the test set predictions should be below 5%, we find this threshold is approximately $\hat{S}_g \approx 0.5$ μM/s (For *E. coli*, the outside signal is the chemical concentration, whose unit is μM.) as shown in Fig 4(B), which is aligns with the theoretical gradient threshold $S_g \approx 0.59$ μM/s. Fig 4(C) displays the SIVM predictions of three different $T_1$. The smaller the $T_1$ is, the larger $s'_{max}$ is. $s'_{max}$ for all three different $T_1$s in Fig 4(C)

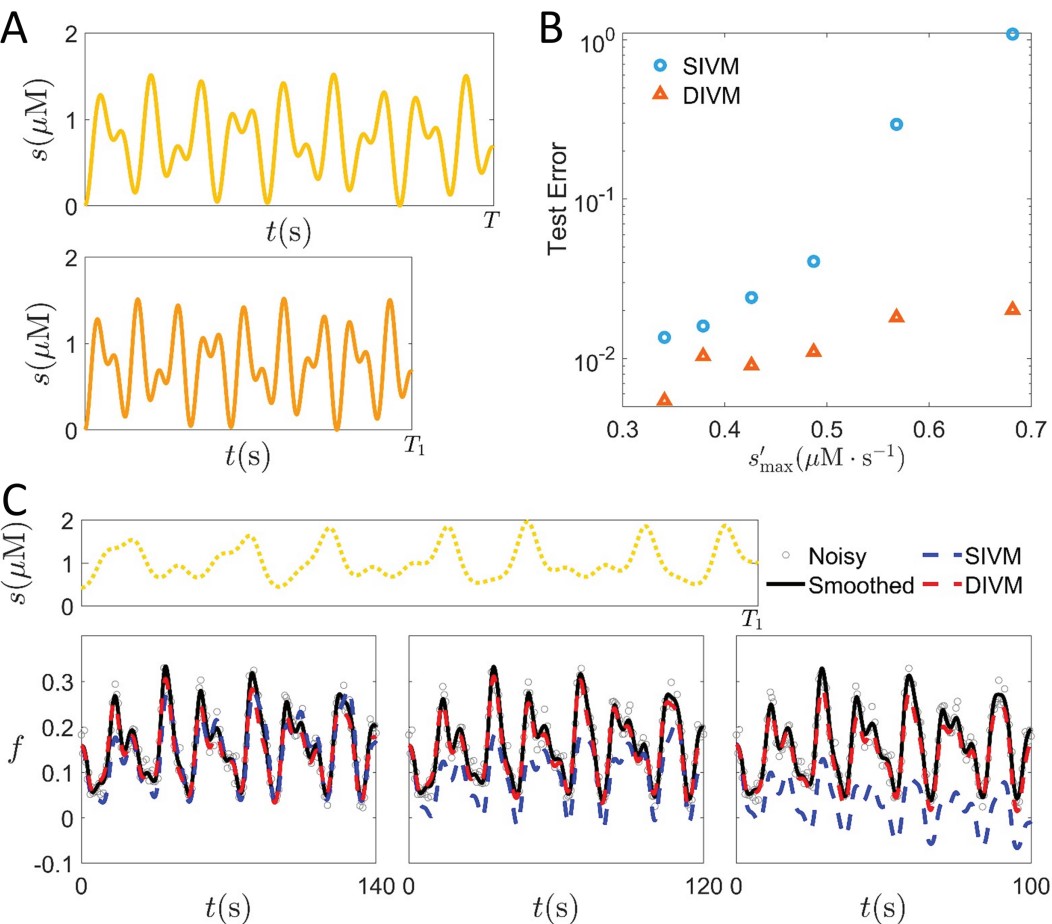

**Fig 4. Identify the _s_ gradient threshold leading to model II degeneration from SIVM and DIVM test errors.**
External stimuli are compressed in time, as shown in (A). We gradually compress the $T = 200$ s stimulus signal to $T_1 = \{180, 160, 140, 120, 100\}$ s. The absolute maximum value of the gradient $s'_{max}$ is correlated with SIVM and DIVM test errors in (B). The error points from left to right correspond to $T_1 = \{200, 180, 160, 140, 120, 100\}$ s respectively. (C) illustrates the SIVM (blue dashed line) and DIVM (red dashed line) prediction outcomes of three experiments with different $T_1$. Here, ELCC stimuli are used in the test set; the subplots from left to right correspond to the three error points for $T_1 = 140$ s, $T_1 = 120$ s, and $T_1 = 100$ s in (B).

are near the threshold. It can be seen that the long-term predictions become worse as $T_1$ decreases.

For the experimental data of _E. gracilis_, as illustrated in Fig 3(D)–3(E) and Table 1, SIVM can model the photo-responsive movement of _E. gracilis_ when $s'_{max} < 4.183 \times 10^{-3}$ W/(m²·s). On the other hand, DIVM can achieve excellent predictive results when $s'_{max}$ becomes larger. Based on the discussions of the degeneration condition of DIVM, this implies that the pathways of _E. gracilis_ photo-response can be modeled by one internal variable when $s'_{max} < 4.183 \times 10^{-3}$ W/(m²·s), while when $s'_{max}$ reaches $10^{-2}$ W/(m²·s), one has to use two internal variables to get the correct behavior. One may set $\hat{S}_g$ to be $4.183 \times 10^{-3}$ W/(m²·s), but to get a better estimate of $S_g$, more carefully designed experimental data are required.

One can further study the extrapolation capability of the algorithms, specifically the prediction accuracy when the value range and gradient range of test-set stimuli exceed those

of the training set. We generate response data using Model II under modified stimulus signals and construct noiseless training and test datasets. Two different categories of test sets are constructed, which are associated with a relative maximum deviation of the absolute stimulus value $R_s$ or a relative maximum deviation of the absolute stimulus gradient $R_{s'}$ from the training set. The detailed method for constructing the two different categories of test sets is provided in Appendix F in S1 Text.

After training the neural networks using both SIVM and DIVM, we apply them to the two different categories of test sets described above. SIVM and DIVM test errors are correlated with the values of $R_s$ and $R_{s'}$ of different data sets, as shown in Fig C(B) and Fig D(B) in S1 Text. As the test data's deviation from the training range increases (both in value and the gradient of stimuli), the test errors of both algorithms grow. When $R_s = 0.49$ ($R_{s'} = 1.38$), SIVM's (DIVM's) error exceeds 10%. Regarding extrapolation capability, our algorithms maintain accurate predictions even when the gradient range expansion extends significantly beyond the training set. DIVM consistently outperforms SIVM in handling value range expansion.

## Design principles for stimuli

Most studies of signal processing and behavioral response are based on measuring responses to simple controllable stimuli. Recent biological experiments have measured behavioral response under varied external stimuli, including step functions (*E. coli* [71], algae [21,34, 65]), ramp [65], exponential ramp [53], sine-wave [34,65,72], and exponentiated sine-wave stimuli with low/high frequency [53]. Inspired by these experiments, we design the above three types of stimulus signals $s(t)$ for training dataset generation.

We further investigate the three types of stimulus signals used in the experiments to explore the impact of different signal combinations on NN's training and predictive accuracy. The first type is the PWC, which is widely used to study the response behavior of unicellular organisms to sudden signal changes [46,47,49,51]. This signal has a zero first derivative almost everywhere except at the jump points. The second type is called the PWCL, where piecewise constant functions are connected by linear segments whose slopes may vary in different transition intervals. This signal has a zero-second derivative almost everywhere except at the turning points. The third type is the LCC, one of the classic smooth signals. The exact mathematical definitions for these three types of signals have been provided in the previous subsections.

For PWC signal, since at almost everywhere $s'(t) = 0$, $G_{\theta_2}^{NN}(f,s)$, $G_{\theta_3}^{NN}(f,s,f',s')$ and $G_{\theta_4}^{NN}(f,s,f',s')$ are untrained. While for PWCL signal, at almost everywhere $s''(t) = 0$, $G_{\theta_4}^{NN}(f,s,f',s')$ is untrained.

Our algorithms are applied to the *E. coli* chemotaxis Model I, II, and III. For the three different types of signals PWC, PWCL, and LCC, the training set comprises one, two, or three of them, as shown in Table 2. $N^{\text{data}} = 20,020$ stimulus-response pairs are used, with 70% allocated to the training set and the rest for testing.

For the SIVM, it is evident that when the training dataset contains only PWC signals, the neural network $G_{\theta_2}^{NN}(f,s)$ cannot be effectively trained. After training the neural network with datasets composed of different signal combinations, while keeping the number of data points and iterations the same, we evaluated the neural network's performance on the same test set. The results show that using a combination of PWC and LCC signals as the training set achieves the best outcome while using only LCC signals as the training set results in the largest errors.

**Table 2. The accuracy of the test set defined in (23) for different combinations of basic stimulus.**

| Training set | Test set | SIVM $E_{\text{test}}$ (%) | DIVM $E_{\text{test}}$ (%) |
|---|---|---|---|
| PWC | PWC, PWCL, LCC | $G_{\theta_2}^{NN}(f,s)$ untrained | $G_{\theta_4}^{NN}(f,s,f',s')$ untrained |
| PWCL | | I (2.42%) | $G_{\theta_4}^{NN}(f,s,f',s')$ untrained |
| PWC, PWCL | | I (2.61%) | $G_{\theta_4}^{NN}(f,s,f',s')$ untrained |
| LCC | | I (3.51%) | II (0.51%), III (2.72%) |
| PWC, LCC | | I (1.90%) | II (0.04%), III (1.09%) |
| PWCL, LCC | | I (2.51%) | II (0.09%), III (0.84%) |
| PWC, PWCL, LCC | | I (2.04%) | II (0.02%), III (0.72%) |

Similarly, within the DIVM framework, we conduct experiments using Models II and III. From Table 2, LCC signals are necessary for the training set; without them, the neural network $G_{\theta_4}^{NN}(f,s,f',s')$ cannot be trained and its predictive performance on the test set is very poor. The first and second derivatives of LCC signals offer abundant information, significantly enhancing the training efficacy of the networks. Among the three different signal combinations, when the training data size is the same, combining LCC signals with either PWC or PWCL signals significantly improves predictive accuracy, which is much better than using LCC signals alone. Utilizing all three types in the training set can improve predictive accuracy, but the enhancement is not significant, as shown in the last row of Table 2.

## Discussion

Microorganisms adjust their motion behavior in response to external signal stimuli, thereby better adapting to environmental conditions. Traditionally, these dynamic changes are described by a system of ODEs for intracellular pathways and a nonlinear response function. However, measuring the temporal dynamics of internal states is challenging, which complicates model construction.

Hence, there is a need to construct models that directly relate external signal stimuli to behavioral responses, without explicit information of the intracellular signaling pathways. This study introduces a novel neural-network framework based on external stimulus-response data. The approach relies on applying the chain rule to eliminate internal variables that simulate intracellular reactions, thereby establishing functional relationships (7) and (14) between the time derivative of stimulus intensity and tumbling behavior. These relationships are then defined as the losses (16) and (17) for training the neural networks.

We have applied this algorithm to various *E. coli* chemotaxis models and experimental data of *E. gracilis*, validating its effectiveness in predicting motion responses under complex environmental conditions. Moreover, the algorithm can infer the potential structure of the response pathways, as has been realized using *E. gracilis* experimental data.

The framework introduced here does not require measurements of intracellular protein levels or knowledge of the specific forms of equations governing intracellular chemical reactions. Instead, it infers the potential structure of response pathways directly from movement data. We employ neural networks to learn representations, enabling the algorithm to generalize to other systems with unknown latent variables. Thus, we claim that this algorithm is a general-purpose, easy-to-implement, and accurate simulator for identifying the governing equations of run-and-tumble dynamics.

The current work is a first step in adapting neural network framework to identify governing equations from stimulus-response data. The two cell types in this study exhibit a similar run-and-tumble movement pattern, thus, the behavioral response data comprise

tumbling fraction changes under different stimuli. Given that our model is derived in a general form, it should, in principle, be applicable to any input-output datasets. Therefore, future work could explore extending this approach to other types of response data to assess its broader applicability.

We focus exclusively on the two cases $J = 1$ and $J = 2$ in this work. Nevertheless, extending the method to systems with larger $J$ requires information about higher-order derivatives of the stimulus and response data, which can be challenging when dealing with noisy experimental data. Moreover, since the time derivative is used in the model, only continuous-in-time data are considered here (but the time sequence can be short); extending the method to discontinuous stimulus and response data is necessary. Notably, several neural network architectures, such as RNNs and Neural CDEs, remain viable for processing long time series and performing prediction tasks. A promising future direction involves integrating our current algorithm with other neural network approaches, which may yield novel algorithms with biological interpretability, noise resistance, stable long-term prediction capabilities, and enhanced applicability to more complex systems.

## Supporting information

**S1 Text. The supplementary document provides Appendix A–F and three supplementary figures for the main text.**
(PDF)

## Author contributions

**Conceptualization:** Yuan Li, Hepeng Zhang, Min Tang.

**Data curation:** Shicong Lei, Yuan Li, Hepeng Zhang.

**Formal analysis:** Shicong Lei, Min Tang.

**Methodology:** Shicong Lei, Zheng Ma, Hepeng Zhang, Min Tang.

**Project administration:** Min Tang.

**Software:** Shicong Lei.

**Supervision:** Zheng Ma, Hepeng Zhang, Min Tang.

**Validation:** Shicong Lei, Yuan Li, Hepeng Zhang, Min Tang.

**Visualization:** Shicong Lei, Yuan Li.

**Writing – original draft:** Shicong Lei, Yuan Li.

**Writing – review & editing:** Shicong Lei, Yuan Li, Zheng Ma, Hepeng Zhang, Min Tang.

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
