## [Decision Letter · Decision Letter 0]

31 Mar 2025

PCOMPBIOL-D-25-00309

Identification of the governing equation of stimulus-response data for run-and-tumble dynamics

PLOS Computational Biology

Dear Dr. Tang,

Thank you very much for submitting your article "Identification of the governing equation of stimulus-response data for run-and-tumble dynamics" to PLoS Computational Biology. After careful consideration, we feel that it has merit and we invite you to submit a revised version of the manuscript that addresses the points raised during the review process.

As with all papers reviewed by the journal, your manuscript was reviewed by several independent reviewers. In light of the reviews (below this email), we would like to invite the resubmission of a revised version that takes into account the reviewers' comments.

The reviewers found the work interesting but raised several concerns, most importantly regarding manuscript organization, and improving on clarification and novelty of the method.

We cannot make any decision about publication until we have seen the revised manuscript and your response to the reviewers' comments. Your revised manuscript is also likely to be sent to reviewers for further evaluation. 

Please submit your revised manuscript within 60 days May 31 2025 11:59PM. If you will need more time than this to complete your revisions, please reply to this message or contact the journal office at ploscompbiol@plos.org. Please include the following items when submitting your revised manuscript:

We look forward to receiving your revised manuscript.

Kind regards,

Calina Copos, Ph.D.

Academic Editor

PLOS Computational Biology

Feilim Mac Gabhann

Editor-in-Chief

PLOS Computational Biology

**Additional Editor Comments (if provided):**

**Journal Requirements:**

**Reviewers' comments:**

Reviewer's Responses to Questions

**Comments to the Authors:**

Reviewer #1: This manuscript presents a neural network-based approach for identifying governing equations of run-and-tumble dynamics without explicitly reconstructing the underlying ODEs for intracellular chemical reactions. The methodology is mathematically sound and provides an interesting perspective on bacterial motility research. While the work has its merits, several areas require clarification and improvement before publication.

1. The study builds on the well-known dynamics of E. coli chemotaxis, with Model I focusing on the slow dynamics of m, Model II incorporating the slow dynamics of m and the fast dynamics of Yp, and Model III including the slow dynamics of both m and FliM. Given the differences in timescale separation among the intermediate variables in the three models, the performance of SIVM and DIVM on the generated data is not surprising. The potentially novel insight is that this method could possibly extract information about the internal dynamics of Euglena gracilis phototaxis. For example, the varying performance of SIVM and DIVM at different s'max values may suggest that the phototaxis pathway can also be described by two variables—one fast and one slow. The authors should expand on this point and connect it to what is known about the phototaxis pathway in Euglena gracilis.

2. The assumption of monotonic dependence of response functions (F) on internal variables (m, n) is critical for inverting the system. While this assumption holds for the well-studied E. coli chemotaxis pathway, no validation is provided for experimental systems like E. gracilis phototaxis. The authors should include sensitivity analyses or empirical evidence to justify this assumption. If monotonicity cannot be verified, alternative approaches should be discussed.

3. The training data rely on synthetic stimuli (PWC, PWCL, LCC). However, real-world signals may exhibit more complex patterns (e.g., stochastic fluctuations). The paper should discuss the framework's robustness to non-synthetic stimuli and whether extrapolation beyond the training ranges (e.g., higher s'max) is feasible.

4. A critical value of the stimulus signal gradient, 0.5 μM/s, is identified, beyond which SIVM becomes inapplicable. However, this specific value lacks meaningful context without further discussion. The authors should relate this value to the parameter values used in the chemotaxis model to provide deeper insight.

5. Citations for the parameter values in the supplemental tables should be provided.

Minor:

Line 241: Define "PWC" upon its first occurrence.

Reviewer #2: The review is uploaded as an attachment.

**Have the authors made all data and (if applicable) computational code underlying the findings in their manuscript fully available?**

Reviewer #1: Yes

Reviewer #2: None

PLOS authors have the option to publish the peer review history of their article (what does this mean?). If published, this will include your full peer review and any attached files.

Reviewer #1: No

Reviewer #2: No

**Figure resubmission:**

**Reproducibility:**

 

---

## [Decision Letter · Decision Letter 1]

30 Jun 2025

Dear dr. Tang,

We are pleased to inform you that your manuscript 'Identification of the governing equation of stimulus-response data for run-and-tumble dynamics' has been provisionally accepted for publication in PLOS Computational Biology.

Best regards,

Calina Copos, Ph.D.

Academic Editor

PLOS Computational Biology

Feilim Mac Gabhann

Editor-in-Chief

PLOS Computational Biology

Reviewer's Responses to Questions

**Comments to the Authors:**

Reviewer #1: The revision addressed my comments, thus its acceptance is recommended.

Reviewer #2: I thank the Authors for their consideration of my comments and those of the other reviewer.

I believe that the requested changes have been made and that the answers provided clarify exhaustively.

I compliment the Authors on the valuable work of reviewing the manuscript and I have no further comments to submit.

**Have the authors made all data and (if applicable) computational code underlying the findings in their manuscript fully available?**

Reviewer #1: Yes

Reviewer #2: None

PLOS authors have the option to publish the peer review history of their article (what does this mean?). If published, this will include your full peer review and any attached files.

Reviewer #1: No

Reviewer #2: No

---

## [Editor Report · Acceptance letter]

PCOMPBIOL-D-25-00309R1

Identification of the governing equation of stimulus-response data for run-and-tumble dynamics

Dear Dr Tang,

I am pleased to inform you that your manuscript has been formally accepted for publication in PLOS Computational Biology. Your manuscript is now with our production department and you will be notified of the publication date in due course.

With kind regards,

Zsofia Freund
